# Multi- class classification of breast cancer abnormalities using Deep Convolutional Neural Network (CNN)

Maleika Heenaye-Mamode Khan[1]*, Nazmeen Boodoo-Jahangeer[1], Wasiimah Dullull[1], Shaista Nathire[1], Xiaohong Gao[2], G. R. Sinha[3], Kapil Kumar Nagwanshi[4]

1 Department of Software and Information Systems, University of Mauritius, Reduit, Mauritius, 2 Department of Computer Science, Middlesex University, London, England, United Kingdom, 3 Department of Electronics and Communication Engineering, Myanmar Institute of Information Technology (MIIT) Mandalay, Myanmar, 4 Department of Computer Science and Engineering, Amity University Rajasthan, Jaipur, India

ʘ These authors contributed equally to this work.

* m.mamodekhan@uom.ac.mu

## Abstract

The real cause of breast cancer is very challenging to determine and therefore early detection of the disease is necessary for reducing the death rate due to risks of breast cancer. Early detection of cancer boosts increasing the survival chance up to 8%. Primarily, breast images emanating from mammograms, X-Rays or MRI are analyzed by radiologists to detect abnormalities. However, even experienced radiologists face problems in identifying features like micro-calcifications, lumps and masses, leading to high false positive and high false negative. Recent advancement in image processing and deep learning create some hopes in devising more enhanced applications that can be used for the early detection of breast cancer. In this work, we have developed a Deep Convolutional Neural Network (CNN) to segment and classify the various types of breast abnormalities, such as calcifications, masses, asymmetry and carcinomas, unlike existing research work, which mainly classified the cancer into benign and malignant, leading to improved disease management. Firstly, a transfer learning was carried out on our dataset using the pre-trained model ResNet50. Along similar lines, we have developed an enhanced deep learning model, in which learning rate is considered as one of the most important attributes while training the neural network. The learning rate is set adaptively in our proposed model based on changes in error curves during the learning process involved. The proposed deep learning model has achieved a performance of 88% in the classification of these four types of breast cancer abnormalities such as, masses, calcifications, carcinomas and asymmetry mammograms.

## Introduction

Breast cancer is known to be the second leading cause of death among women in the world. Breast cancer occurs when there is an abnormal growth of a mass of tissues of malignant cells.

**Data Availability Statement:** All relevant data are within the manuscript and its Supporting Information files.

**Funding:** The author(s) received no specific funding for this work.

**Competing interests:** The authors have declared that no competing interests exist.

These cells originate from the milk glands of the breast. The classification of these cells is determined by the rate at which these unusual cells progress and the effect that they have on other normal cells that can eventually affect the whole body [1]. According to statistics of the World Health Organization (WHO), breast cancer is the most common type of cancer occurring in women globally, accounting for approximately 2.1 million new breast cancer cases [2]. In 2018, it is estimated that 627,000 women died from breast cancer–that is approximately 15% of all cancer deaths among women. There are research studies and work in the literature on detection and classification of breast cancers using some conventional methods [3–6] that have not used any specific type of machine learning.

It is reported that breast cancer is detected if specific symptoms appear. However, it was found that many women who have breast cancer do not have any symptoms. Primary prevention of breast cancer is not yet a reality except by undergoing prophylactic mastectomy for those who have the BRCA1 or BRCA2 gene mutation, known to cause breast cancer [7]. The effect of treatments administered to patients can also have an effect on the prognosis of the disease and the possibility of relapse in cancer patients. Surgery, chemotherapy, radiotherapy and hormone therapy are among the classical medical treatments used on breast cancer patients. Thus, screening is recommended as mandatory step so that the detection of breast cancer be made at early stage. If the cancer is detected at early stage then the lives can be saved [4]. Early detection of breast cancer helps in the early diagnosis and treatment. This is vital because prognosis is very important for long-term survival [7, 8]. Delays in the treatment of cancer contributes to the propagation of the illness and causes delays in the treatment. According to a study conducted by [9], it was noticed that those who have started their treatment within 3 months from the appearance of breast cancer symptoms have higher chance of survival and can reduce the proliferation of malignant cells from the body.

Advances in image processing especially medical image processing are bringing hopes in devising appropriate automated breast cancer detection and classification applications. The algorithms being deployed in deep learning have the capabilities of using layers of the neural networks to recognize patterns; and the computer algorithms are becoming very useful in the medical field. Despite continuous research in automated breast cancer applications, the correct identification or classification of breast abnormalities is still a challenge. In addition, deep learning requires large training data, which is very difficult to obtain in the medical domain. Thus, there are scopes in the further exploration of automated breast cancer detection applications to improve the accuracy of cancer screening. We have used a deep learning model for classification of abnormalities in this work. The main contribution of this research is summarized as follows:

1. We have applied a pre-trained model ResNet50 and have devised ways to overcome overfitting models

2. We have developed an enhanced deep CNN model by varying the learning rate, which is an important parameter, that influence performance. We have also considered ways of using the learning rate in adaptive manner during training. Our model has been able to differentiate between various types of abnormalities that cause breast cancer.

## Related work

Considered as a screening procedure, breast self-examination is done by the individual themselves. By palming the breasts in different angles and under different pressures it is possible to detect any difference or changes in the breasts. However, the breast examination is the least reliable way to detect cancer. Mammography has emerged as an alternative and is being widely

used in the medical field. However, relying only on mammograms has a high risk of false positives which often lead to unnecessary biopsies and surgeries [1].

With the development of automated medical applications, researchers have been developing automated breast cancer detection systems. Many applications have been deployed using machine learning techniques. In [1], the use of machine learning has been reported to determine the types of treatment that need to be administered to cancer patients. It has been reported by several authors [10–12] that there has not been any significant improvement in accuracy with the early computer aided applications for breast cancer detection that were developed. In [13], GoogleTensorFlow is used to implement machine learning algorithms, that were applied on the breast features for classification. The Wisconsin datasets were used, where 569 data points were considered, out of which 212 malignant cases and 357 benign cases were seen. Features that were used were mainly the details of the nuclei of the image which were directly computed from the digital image obtained from the fine needle aspirate (FNA) of the masses from breast images. The mainly used features are smoothness, concavity, texture, compactness, perimeter, fractal metrics and few more interrelated attributes. The support vector machine (SVM) and Gated Recurrent Unit (GRU) were utilized. The GRU is actually a variant of recurrent neural network (RNN). The networks were used for binary classification in which training and testing were performed. The dataset includes 30% of testing data and 70% of training dataset. In the training process, a number of metrics were used such as true positive rate (TPR), number of epochs, accuracy, data points, true positive rate (TPR), false negative rate (FNR) and false positive rate (FPR). The proposed machine learning method provided classification performance of 93.75%. In addition, K-fold cross validation was also used to confirm the validity of the results.

In [14], the authors have used the gray level co-occurrence matrix (GLCM) to extract the statistical texture features from mammograms namely: Entropy, Angular Second Moment, Contrast, Mean and Difference Moment from each density function. In [1], numerous ways were studied to detect breast cancer while emphasizing the use of machine learning. A hybrid model was proposed with a combination of SVM, Artificial Neural Network (ANN), K Nearest Neighbour (KNN), and Decision Tree (DT) algorithms. With varying combinations of these techniques, SVM was found as the most popular method that resulted an accuracy of 99.8% irrespective of the fact that it was used alone or as a hybrid. On the other hand, authors in [15] used thresholding techniques to segment the out boundaries of the breast images and applied the Discrete Wavelet Transform (DWT) to extract the features. The success of this method has been noted to be of a rate higher than 90%.

In another work reported in [16], histopathological images were used to detect breast cancer. Histopathological images show observations obtained from biopsy. These images contain both local and hidden patterns. In order to determine the hidden patterns, unsupervised techniques namely Convolutional Neural Network (CNN), a Long-Short-Term-Memory (LSTM), and a combination of the CNN and LSTM models. SVM was then used to classify the images. In [17], authors have used a feed-forward back propagation network having number of hidden layers as 3. In these hidden layers, number of neurons were 50, 10 and 50 respectively in first, second and third hidden layers. In training process, suitable features were extracted which were used for diagnosis of the images. In this research, individual stage of cancer was linked through the neural network. In [18], authors worked on a CAD to distinguish between the different regions of breast tissues. Using both the standard MIAS and DDSM database, they enhanced the peripheral region of the mammograms and extracted 422 features from the ROI. They used selection techniques such as t-test and Sequential Backward selection. The four classifiers: K Nearest- Neighbour (KNN), Support Vector Machine (SVM), Linear Discriminant Analysis (LDA) and Quadratic discriminant analysis (QDA) were applied and compared.

Despite the ongoing development of machine learning techniques, there are no significant improvements in the performance of these applications. In the meantime, deep learning, which learns representations from data which promotes the learning of successive layers of increasingly meaningful representations, have been successful in visual object detection and classification in many domains. Eventually there is a growing interest in the exploration of deep learning techniques for breast cancer detection applications. Recent studies on the application of deep learning on breast cancer have provided good preliminary output and showed scopes for further exploration [19, 20]. Despite the numerous research and application of neural network, none of the work developed so far could classify the breast abnormalities, clearly showing a research gap in this area.

## Material and methods

This section highlights about the methodology used in implementing the deep learning for classification.

### Architecture of the proposed solution

Fig 1 shows various components of the proposed automated breast cancer detection and its application. A customized dataset is firstly constructed and then the images are pre-processed to remove blurs and noises. A deep convolution neural network has been developed in addition to the application of an existing pre-trained deep learning model. The enhanced model has been tested and evaluated.

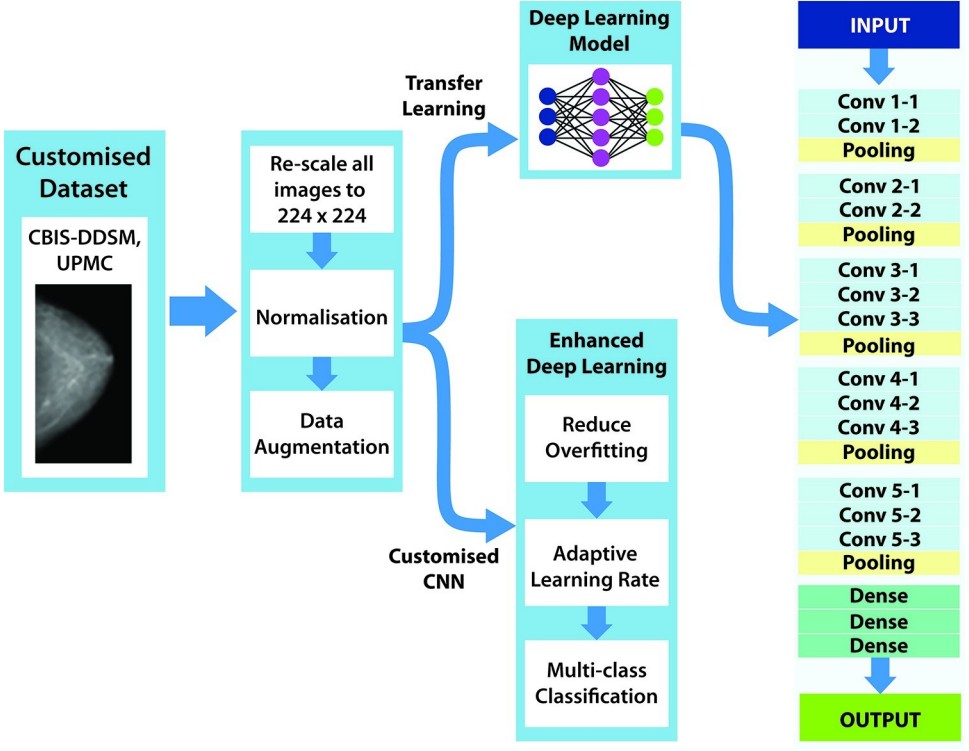

**Fig 1. Architecture of the proposed model.**

## Dataset preparation

In this research work, several abnormalities that potentially determine breast cancer are considered. The main concern is that the available online datasets do not cater for all the abnormalities that can occur in the breast. Thus, a customized dataset using CBIS-DDSM (available at: https://wiki.cancerimagingarchive.net/display/Public/CBIS-DDSM) and UPMC (available at: https://cancerregistrynetwork.upmc.com/upmc-network-cancer-registry/standardized-data-set-2/) has been created to contain images having masses, calcifications, carcinomas and asymmetry mammograms. The DDSM is a publicly available dataset that has been created by South Florida University and the classification of the breast abnormality has been done by expert radiologists. The dataset was created for research-based applications and studies for breast cancer detection. The computer-aided diagnosis (CAD) operates over the dataset. The image dimension of the breast images was taken as 3000 X 4800 pixels having the resolution up to 42 microns and depth of 16 bits. Around 2620 mammograms were included in the DDSM dataset which were scanned film images. In each of the cases, 4 different vies of breast images were taken so that the classification is satisfactory. There were two Mediolateral Oblique (MLO) views in each of the breast images and other two were Cranio-Caudal views. The CBIS-DDSM dataset is a subset of the DDSM one and it consists of 6775 studies. This subset is however curated by a trained mammography expert. Thus, it is recognized as an updated and standardized version of DDSM. The format of the images in DICOM is a decompressed version of the images and represents abnormalities like calcifications, mass, asymmetry and carcinomas. The UPMC is a dataset which contains tomo-synthesis images, where asymmetry breast abnormalities and mass images were selected. Fig 2 shows one sample of each of the different types of abnormalities considered namely: asymmetry, calcification, carcinoma and mass.

## Data pre-processing

This step is used to enhance the quality of the image and increase the chance for a better abnormality classification. The intensity between objects is improved to better highlight the breast tissue and noise introduced at image capture are also eliminated. Several filtering techniques have been analyzed and applied on the images to select the most appropriate one in this

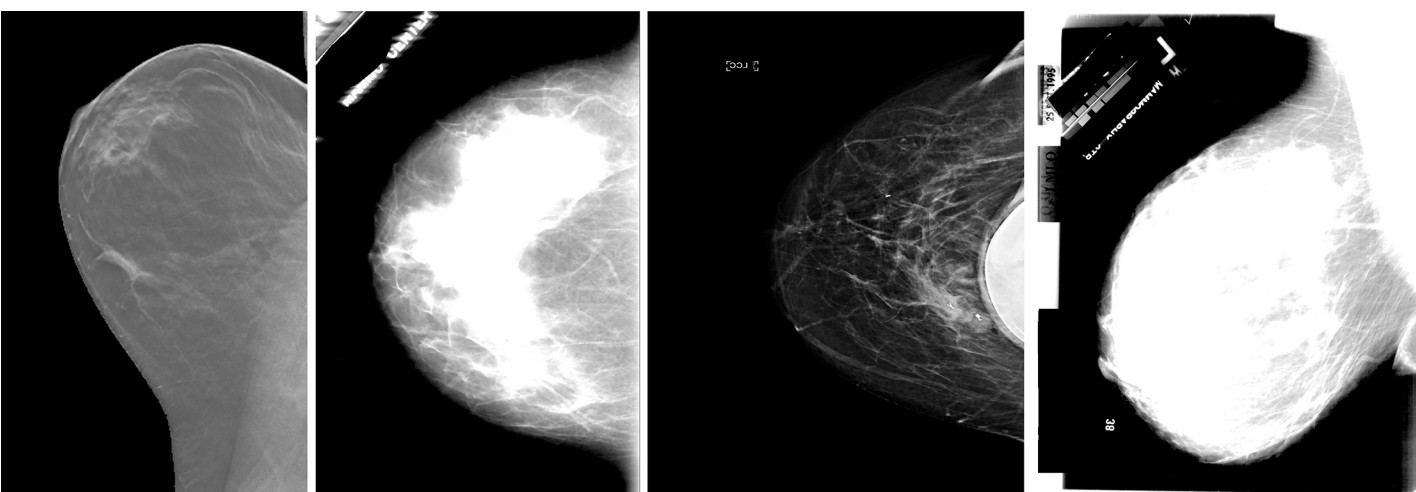

**Fig 2.** (a) Asymmetry (b) Calcifications (c) Carcinoma (d) Mass.

context. One popular filter namely the Wiener filter has been applied which is normally used to curb the amount of noise in the image. The noise signal is attenuated and brought down to some minimum level, rather than complete elimination which is not possible. It eliminates the additive noise and using inverse filtering, we can recover blurred image. However, as discussed in [21], blurry sharp contents or edges of images such as edges, lines and other fine details are messed up after the application of this filter. Median filter is yet another technique that is popularly used as filter technique and works by replacing each pixel value with the centroid or median value of the surrounding neighborhood of the pixel under the rectangular region it is working on. Eventually, it can reduce the amount of intensity variation between one pixel and the other pixels while keeping the sharpness of image edges [22]. A detailed analysis was conducted in [23] to show the complexities of the applications of pre-processing techniques in different types of medical images and the need for this process. In this work, Contrast limited adaptive histogram equalization (CLAHE) has been chosen as it operates on small region of the breast image rather than the entire image and apply equalization on each of them. In addition, the breast images had less broken lines. The images are in Dicom format and has been converted into jpg. Convolution Neural Network (CNN), which is one model that automate the process of feature construction is widely being adopted. Though these models have great potential in analyzing images, large datasets are required for the training. Since there are limited data currently available in the medical field, data augmentation was used to expand the dataset. Eventually the breast images were augmented by a factor of 10 by applying data augmentation techniques. The created dataset has 10,200 images, after applying data augmentation namely flipping, rotation, cropping and translation on each type of abnormalities. The breast images were then segmented from the background to eliminate unwanted background information. Fig 3 shows a pre-processed image using CLAHE.

## Transfer learning using pre-trained ResNet50

Transfer learning is based on a single approach of using already learned features from one task and applying to another task without the need to learn from scratch. This is normally conducted on already built Convolutional Neural Network (CNN) models which have been trained on the most commonly used and well-known datasets [24]. The CNN model analyses an input image and assigns weights to various aspects of the image with the aim of differentiating one image from other images. It consists of several layers, as shown in Fig 4.

CNN consists of multiple sets of convolutional, pooling and fully-connected layers. The convolutional layers use the images to extract its features [25], and then the pooling layers are used for selecting two samples and discard the next one [26]. As a result, less data is sent in the

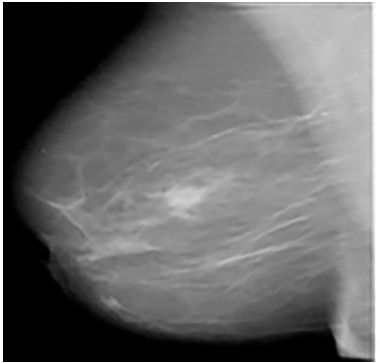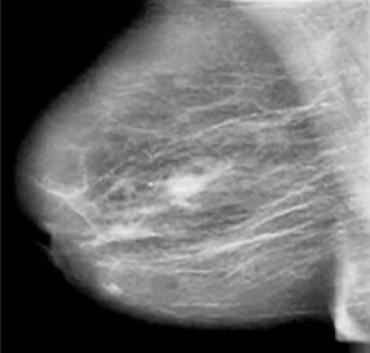

**Fig 3.** (a) Original Image (b) Image after CLAHE.

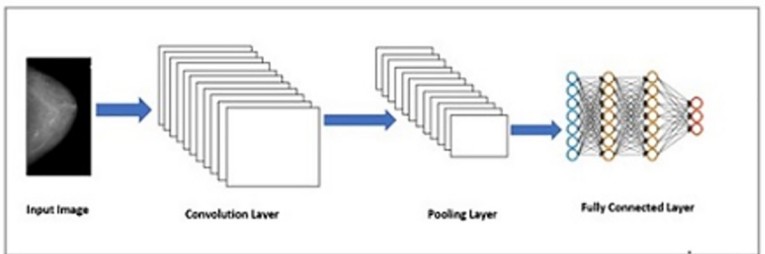

**Fig 4. Layers in CNN.**

layers which reduces the complexity of the computations. The pooling layers have another impact which is it can also initiate translation invariance. The fully connected layers are actually the units of the top and final layers being connected together. We have applied ResNet50, which is an improved version of ResNet model. It consists of 50 layers of shortcuts connections with the residual blocks. The shortcuts make use of less computation and as well as provide rich combination features. ResNet50 consists of one convolutional layer, then a set of normalization layers, and lastly 16 residual modules found in between two pooling layers [27]. ResNet50, which uses 3x3 filters to perform spatial convolution, was introduced for the classification of abnormalities of the breast images. The weights were first set and the convolution base which was used as the feature extractor was frozen. The spatial size of the feature map is reduced by adopting Maxpooling method. Our dataset was split into 60% for the training set and 40% for the testing set.

## Development of the enhanced CNN model

The CNN model is used for training the data very often in medical image diagnosis, analysis and its applications. In fact, the medical imaging in CAD system becomes successful because of CNN use [28, 29]. This substantiates the use of CNN for our application in breast cancer detection, of course with some improvement or enhancement. Generally, a CNN comprises of hundreds of neurons structured in different layers, forming the kernels. The size of kernels is chosen small as the depth of input image has to be same. A receptive field which is actually a small region, is connected with neurons. This is done because it is very difficult to connect all neurons to the previous outputs, especially when the input has high dimensional images. Let us understand by an example: For an image of size 100 x 100 having 10000 pixels, there are 100 neurons which will require or produce 1000000 parameters. Now, each neuron does not need to have weight of full input dimension, rather the neuron will hold the weight corresponding to the kernel dimension since the kernels of small dimension have been connected with neurons. The kernel needs to slide over the image both in height and width so that high level features are extracted and two-dimensional (2D) activation map is produced. The rate of sliding can also be controlled using a suitable parameter. Finally, the output of convolution layer is stacked and activation map defines the input for next layer to be propagated.

Our architecture is based on the Sequential model, which allows the model to stack sequential layers of the network in order from input to output. The convolution layer uses filters that perform convolution operations as it is scanning the input image with respect to its dimensions. Its hyper-parameters include the filter size and stride, which represents the step size between consecutive receptive filters. The resulting output is called feature map or activation map. First 2D convolutional layer is added to process the input breast images. The first argument passed to the convolution layer function is the number of output channels and in our

case, we have used 16 output channels. In our model, we have used 3x3 filter kernel, with stride 1, which slide across the width and height of the image, to perform spatial convolution. During the experiments we have also experience with different kernel size (from 1 to 7) to analyze the differences. We have noticed that the small kernel size 1×1 acts as a bottleneck. We have considered padding so that the input image gets fully covered by the filter and stride specified in this model. We have applied the rectified linear unit (RELU) activation function, which is defined mathematically as: $f(x) = (0,x)$, where the input is represented by $x$. Max pooling is used to reduce the spatial size of the feature map. The Maxpooling function downsamples the input representation by taking the maximum value over the window defined by pool_size for each dimension along the feature's axis. Similarly, the procedure is repeated with two more convolutional layers with 32 and 64 output channels. Initially, we have used a filter of 2x2 max pooling. Inspired from the work conducted in [30] for addressing the degradation problem, we have used the same concept of deep residual learning framework. It works on repeated units of convolution layers having the filter size 1x1,3x3 and 1x1. Global average pooling was used to compute the activation of each feature map.

## Results and performance evaluation

To evaluate the models, the datasets were divided into three sets namely training, validation and testing set. The training set, which represents the sample data, is used to fit the model. The latter learns from this data, which is 60% taken from the dataset. To ensure that the four classes (mass, carcinoma, asymmetry and calcifications) are well represented, 15% from each category were considered. To avoid unbiassed evaluation on the training set, a validation set is chosen to tune the parameters. Out of the remaining 40%, 20% was allocated to the evaluation set and 20% to the test set. The main role of the validation set is to choose the best combination of parameters for our model. As for the test set, it is being used for the final evaluation of the model. Thus, the metrics adopted are training accuracy, validation accuracy and testing accuracy. Initially, during testing, the training accuracy that was achieved was 96.8% and a validation accuracy of 36.7%.

The initial results clearly show that the model is overfitting and is not performing well on generalized datasets. To address this problem, drop-out regularization was considered. In fact, drop-out regularization is known to ignore both hidden and visible neurons which are selected at random during the training process. It removes inputs to a layer which might be the features in the image or activation from a preceding layer. Eventually, this stimulates a different network structure which forces the model to learn more robust features, thus, increasing the generalizing power of the model. The datasets were also re-constructed using the raw data and the augmented images. Initially, attention was not given to the proportion of images for the three classes of images. While experimenting with individual classes, it was seen that there were too few images in one class, and this may be affecting the validation performance. Images were augmented and more images were considered under each class. After drop-down regularization and the generation of the new dataset, we have achieved a training accuracy of 92.8% and validation accuracy of 86.7%. As reported by [31], overfitting hinders the generalization of models. The solutions investigated are early stopping, removal of noise in the data, data augmentation to fine tune parameters and regularization. Likewise, we have investigated the application of drop-out regularization and use augmented datasets to handle overfitting leading to better accuracy.

The model is further enhanced after experimenting with learning rate. After several trials, it was deduced that the model could preserve primitive features of the bottom layers. However, it was observed that more training was required to learn the higher order features. In fact, we

updated the weights of network at each epoch and at certain value of learning rate during training process. It is very important to choose an optimal learning rate to minimize error. One potential solution is to choose a constant learning rate. But still, the system will cause fluctuations in the curve if the value is larger than the optimal value. On the other hand, the model will reduce the convergence rate if the chosen value is less than the optimal value. In addition, choosing an optimal value of learning rate is a challenging task as can be seen in [32]; and thus, we have used two ways to make the learning rate adaptive in entire training process during the experimentation [33]. In first approach, we have considered learning curve which is checked at each step of implementation. If the curve is found to be increasing, then the rate of learning is reduced by multiplication factor which is done by a variable coefficient whose value is less than one. If the increasing trend continues the learning rate is further increased, again by multiplying by a variable coefficient having its value more than one. In the similar manner, if we find constant error curve then we can maintain constant value of rate of learning. In Eq (1), we set the learning rate at each step, in which $\eta(t)$ indicates the learning rate value after doing n$^{\text{th}}$ epoch. The values of variables $\lambda_1(t)$ and $\lambda_2(t)$ respectively in Eq (2) and Eq (3) are coefficients over time that vary at each iteration of training process. The value of $\lambda_1(t)$ is less than one and value of $\lambda_2(t)$ is kept greater than one. This helps to decrease and increase the rate of learning in comparison with previous step used. The coefficients $\alpha_1$ and $\alpha_2$ should be evaluated between the interval $\lambda(0)$ and $\lambda(0) \pm \gamma$, of $\lambda(t)$ value. In this manner, the values are chosen that can also be seen in Eq (4), where 'm' represents the maximum value of epochs used in training; $\gamma$ indicates the interval for choosing the desired changes in $\lambda$ and $\beta$. This assists in determining the speed while the values in the interval tends to decrease. We can clearly see that smaller is the value of $\beta$, faster is the change in value of the $\lambda$ by either increasing or decreasing to the final value.

$$\eta(t + 1) = \{\eta(t) * \lambda_1(t) \; if \; (E(t) > E(t + 1) \quad \eta(t) * \lambda_2(t) \; if \; (E(t) < E(t + 1) \quad \eta(t) \qquad if \; (E(t) = E(t + 1)) \tag{1}$$

$$\lambda_1(t) = \lambda_1(t - 1) - \alpha_1 e^{-\frac{t}{\beta}} \tag{2}$$

$$\lambda_2(t) = \lambda_2(t - 1) + \alpha_2 e^{-\frac{t}{\beta}} \tag{3}$$

$$\alpha e^{-\frac{1}{\beta}} + \alpha e^{-\frac{2}{\beta}} + \cdots + \alpha e^{-\frac{m}{\beta}} = \gamma \tag{4}$$

$$\alpha = \frac{\gamma \left(1 - e^{-\frac{1}{\beta}}\right)}{e^{-\frac{1}{\beta}} \left(1 - e^{-\frac{m}{\beta}}\right)} \tag{5}$$

On the other hand, Adaptive Moment Estimation (ADAM) optimization algorithm has also been applied and analyzed. Adam attempts to compute adaptive learning rates for each parameter. This is very useful in complex network structures because different parts of the network have different sensitivity to weight adjustment. At times, in breast images, the abnormalities cover a very small region and cannot be detected easily. A very sensitive part usually requires a smaller learning rate. Adam maintains two additional variables $m_t$ and $v_t$ for each variable to be trained:

$$m_t = \beta_1 m_{t-1} + (1 - \beta_1)g_t \tag{6}$$

$$v_t = \beta_2 v_{t-1} + (1 - \beta_2)g_t^2 \tag{7}$$

where $t$ represents $t^{\text{th}}$ iteration and $g_t$ is the calculated gradient. $m_t$ and $v_t$ are moving averages of the gradient and the squared gradient. From the statistical perspective, $m_t$ and $v_t$ are values of the estimates of the first and second moment respectively. The first moment here is the mean value and the second moment corresponds to the uncentered variance of the gradients. Now, when the values of $m_t$ and $v_t$ are initialized as vectors of 0's, then these are biased towards 0, especially during the initial steps, and especially when $\beta_1$ and $\beta_2$ are close to 1. To solve this problem, $\hat{m}_t$ and $\hat{v}_t$ are being introduced.

$$\hat{m}_t = \frac{m_t}{1 - \beta_1^t} \tag{8}$$

$$\hat{v}_t = \frac{v_t}{1 - \beta_2^t} \tag{9}$$

ADAM's weight are being updated as follows:

$$w_{t+1} = w_t - \frac{\eta}{\sqrt{\hat{v}_t} + \epsilon} \hat{m}_t \tag{10}$$

In practice, ADAM converges quickly. Other parameters do not need adjustment. To ensure that the performance of the model does not remain stagnant, the ReduceLROnPlateau is being used. It monitors the accuracy of the model and decreases the learning rate by a certain factor once the accuracy has stop improving. The fluctuation was seen in learning rate error curve while using the constant rate in the training. The ADAM optimizer was also adopted by [34] and have achieved positive results.

The refined model was then tested on the testing set. The enhanced model has achieved a training accuracy of 96.7% on the training set and 83.3% on the validation set. This shows that the enhanced CNN performs better and could learn more general features to correctly classify the breast abnormalities. Fig 5 shows the loss on the training data and the loss on the validation data which is converging towards zero.

Actually, the training loss is actually higher than the validation loss. This clearly shows we are controlling overfitting data in our model. The model was then applied to the testing set and the performance is summarized in Table 1.

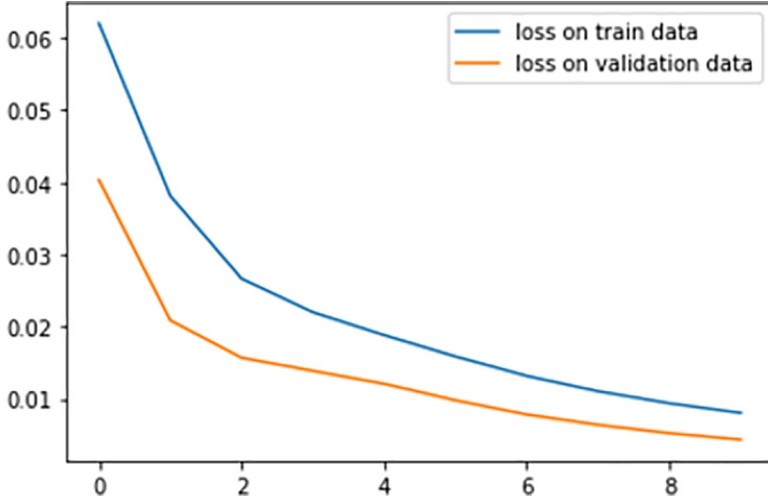

**Fig 5. Loss on training and validation data.**

**Table 1. Performance of the enhanced CNN model.**

| True Labels vs Predicted Labels | Calcifications | Mass | Asymmetry | Carcinomas |
|---|---|---|---|---|
| Calcifications | **0.91** | 0.06 | 0.08 | 0.03 |
| Mass | 0.01 | **0.85** | 0.06 | 0.04 |
| Asymmetry | 0.03 | 0.04 | **0.82** | 0.03 |
| Carcinomas | 0.04 | 0.05 | 0.04 | **0.90** |

An overall performance of 88% was achieved for the multi-class classification of the abnormalities.

The ResNet50 and our proposed model have been refined by minimizing the loss function and fine tuning all the parameters using the training accuracy and validation accuracy. In the testing phase, ResNet50 has achieved an overall accuracy of 81.5% while our enhanced model has reached 88%. Table 2 summarizes the results attained.

## Comparison with existing state-of-the art

Up to our knowledge, no work conducted so far have achieved the four classification of abnormalities: masses, calcifications, carcinomas and asymmetry. In one work, [35] have used CNN to classify carcinoma, which is one type of abnormality used to detect the possibility of breast cancer. The authors have achieved an accuracy of 77.8%. In order to avoid overfitting, the authors have applied data augmentation for the different patches. Likewise, CNN have been trained on histopathological images and fusion of breast images was investigated [36]. In this work, it was reported that CNN performed better compared to hand-crafted techniques. It was concluded that CNN should be further investigated as well as ways to improve the accuracy. We analyzed the genes expressions that helped in detecting and classifying the abnormalities in the breast cancer images using CNN. In line, [37] have explored CNN to classify breast cancer using genes expressions. The results have demonstrated scopes in using CNN. However, the work has not explored multi-class classification which is more challenging task. In a work conducted by [38], CNN was used to classify breast cancer into benign and malignant. To detect masses, feature fusion was applied using CNN deep features. Extreme learning machine (ELM) clustering was then applied to classify masses into benign and malignant. Most of the work was conducted so far focusses on ordinary classification of cancer tissues as either malignant or benign using CNN. The preliminary results provide scopes for the further exploration of CNN. In an analysis conducted by [39], it was reported that there is an urgent need to improve imaging analysis and explore detection techniques using AI for breast cancer detection. Breast cancer is determined by several abnormalities and thus, this work contributes in detecting and classifying these abnormalities. Our work has achieved an overall accuracy of 88%. Table 3 compares our proposed solution with existing research work.

## Conclusion

The diagnosis of breast cancer in an early stage can help in the reduction of the mortality caused by breast cancer. The remarkable success of deep learning and image processing has

**Table 2. Summary of performance of the models developed.**

| Model | Testing Accuracy (Overall) |
|---|---|
| RestNet50 Model | 81.5% |
| Enhanced CNN Model | 88% |

**Table 3. Summary of comparison of proposed model with existing works.**

| Existing Work | Research Conducted | Accuracy | Discussion |
|---|---|---|---|
| Araújo et al. (2017) [35] | Classification of only 1 type of abnormality, that is, carcinoma into normal, *in situ* carcinoma and invasive carcinoma | An Accuracy of 77.8% was achieved for four classes | In this work, only one type of abnormality has been considered. However, for this abnormality, the classification was done on 4 different classes. |
| | | | An Accuracy of 77.8% was achieved for four class |
| Spanhol et al (2016) [36] | Investigation on the different layers of CNN to investigate on the parameters and performance | An accuracy of around 85% for the fusion of images | In this work, the different layers of CNN were analyzed. Fusion was done by combining the different patches results with the whole image. It was mentioned that more research should be carried out to determine ways of achieving better accuracy. |
| Elbashir et al. (2019) [37] | Classification of breast cancer using genes expressions. | An accuracy of 98.76% to determine whether a breast is cancerous or not | This work used the gene expressions to classify cancers. Research have shown that CNN is a potential technique that can be used to investigate patterns. However, the work has not explored multi-class classification which is more challenging. |
| Wang et al. (2019) [38] | Detection of masses using CNN through feature fusion | An overall accuracy of 87% | CNN was used to detect masses. The ELM was used to classify masses into benign and malignant. Multi-class classification was not investigated |
| Proposed Work | Detection and Classification of breast cancer abnormalities | An overall accuracy of 88% | In our work, we have been able to detect masses, calcifications, carcinomas and asymmetry mammograms. The detection of these abnormalities helps in the early detection and eventually diagnosis of breast cancer |

spurred the development of automated medical applications. There are various abnormalities in the breast like carcinomas, masses, lumps, calcification and asymmetry, which can indicate a potential breast cancer. A deep learning model has been developed to help the radiologist in reading a breast image. Firstly, a pre-trained model, namely the Resnet 50 has been applied to our datasets, created from CBIS-DDSM and UPMC. Then, data Augmentation was applied on the datasets to obtain a varied set of images. To further improve the model, an enhanced CNN was formulated by monitoring the learning rate. It was deduced that constant learning rate does not allow the model to learn new parameters. In the initial testing phase, the model shows that it was overfitting and thus, the parameters were improved. Our enhanced model achieved a recognition rate of 88% for the multiclass classification of the various abnormalities namely mass, calcification, asymmetry and carcinomas. More breast image datasets taken under varied conditions should be made available to the research community to improve the diagnosis of breast cancer. Followed by image processing, artificial intelligence techniques can be applied to determine the most appropriate imaging technology to be adopted. Alongside, techniques should be improved further to detect and classify the abnormality in the breast at an early stage, thus, determining the most appropriate treatment to be administered to the patient to reduce deaths caused by breast cancer.

## Acknowledgments

We would like to thank Dr Sunilduth Baichoo and Mr Somveer Kishnah for their valuable feedback.

## Author Contributions

**Conceptualization:** Maleika Heenaye-Mamode Khan.

**Data curation:** Nazmeen Boodoo-Jahangeer.

**Formal analysis:** Maleika Heenaye-Mamode Khan, Nazmeen Boodoo-Jahangeer, Wasiimah Dullull, Shaista Nathire, Xiaohong Gao, G. R. Sinha, Kapil Kumar Nagwanshi.

**Funding acquisition:** Maleika Heenaye-Mamode Khan.

**Investigation:** Maleika Heenaye-Mamode Khan, Wasiimah Dullull, Shaista Nathire.

**Methodology:** Maleika Heenaye-Mamode Khan, Xiaohong Gao, Kapil Kumar Nagwanshi.

**Project administration:** Maleika Heenaye-Mamode Khan.

**Resources:** Nazmeen Boodoo-Jahangeer, G. R. Sinha, Kapil Kumar Nagwanshi.

**Software:** Nazmeen Boodoo-Jahangeer, Wasiimah Dullull, Shaista Nathire.

**Supervision:** Maleika Heenaye-Mamode Khan.

**Validation:** Xiaohong Gao, G. R. Sinha.

**Visualization:** Maleika Heenaye-Mamode Khan, Nazmeen Boodoo-Jahangeer.

**Writing – original draft:** Maleika Heenaye-Mamode Khan.

**Writing – review & editing:** Maleika Heenaye-Mamode Khan, Nazmeen Boodoo-Jahangeer, Wasiimah Dullull, Shaista Nathire, Xiaohong Gao, G. R. Sinha, Kapil Kumar Nagwanshi.

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
