## [Decision Letter · Decision Letter 0]

13 Jul 2021

PONE-D-21-18653

Multi- class classification of breast cancer abnormalities using Deep convolutional neural network (CNN)

PLOS ONE

Dear Dr. Heenaye- Mamode Khan,

Thank you for submitting your manuscript to PLOS ONE. After careful consideration, we feel that it has merit but does not fully meet PLOS ONE’s publication criteria as it currently stands. Therefore, we invite you to submit a revised version of the manuscript that addresses the points raised during the review process.

The manuscript had been reviewed by 2 reviewers. Reviewer 1 was of the view that manuscript partly describes a technically sound piece of scientific research and recommended major revision. Reviewer 2 was of the opinion that your manuscript describes a technically piece of scientific research, however, he had made certain observations and also recommended major revision. 

After thorough consideration of comments of Reviewer 1 and Reviewer 2, my decision is "major revision". Please incorporate comments raised by both reviewers. 

We look forward to receiving your revised manuscript.

Kind regards,

Gulistan Raja

Academic Editor

PLOS ONE

Journal Requirements:

2. Please amend your Methods section to provide links to all of the online data sets used in the study.

Reviewers' comments:

Reviewer's Responses to Questions

**Comments to the Author**

1. Is the manuscript technically sound, and do the data support the conclusions?

Reviewer #1: Partly

Reviewer #2: Yes

2. Has the statistical analysis been performed appropriately and rigorously? 

Reviewer #1: Yes

Reviewer #2: Yes

3. Have the authors made all data underlying the findings in their manuscript fully available?

Reviewer #1: Yes

Reviewer #2: Yes

4. Is the manuscript presented in an intelligible fashion and written in standard English?

Reviewer #1: Yes

Reviewer #2: Yes

5. Review Comments to the Author

Reviewer #1: The research paper contributes towards classification of breast cancer abnormalities using Deep convolutional neural network. The four types of breast cancer abnormalities considered in the work are masses, calcifications, carcinomas and asymmetry mammograms.

Introduction part is well written. Introduction/Related work must conclude with clear research gap.

Data set description is clearly mentioned in the proposed work. Authors have used CLAHE for image pre-processing. Need to analyze this image pre-processing technique on various parameters, namely, Illumination corrections, Blur and focus corrections, Filtering and noise removal, Thresholding Edge enhancements, Morphology, Segmentation, Region processing and filters, Point processing, Color space conversions (Whichever is appropriate for given data set).

Proposed architecture is well defined.

Results section need to be improvised. Authors may implement classification using other deep learning models and compare results in terms of validation accuracy and loss, with enhanced CNN model.

Authors are suggested to include more analysis graphs in the results section. Authors may analyze the output by considering various epochs in the training/testing phase. Authors need to mention various parameters used during the implementation of CNN. The typical parameters could be Activation Function, Optimizer, Loss function, Metric (Whichever is suitable for proposed work).

Authors are suggested to include Future work in this area.

Overall evaluation:

The researchers have attempted to give a solution to the classification of breast cancer abnormalities using Deep convolutional neural network. The paper needs a revision by incorporating the changes mentioned above before it is accepted for publication.

Reviewer #2: In this work, author have developed a Deep Convolutional Neural Network (CNN) to segment and classify the various types of breast abnormalities, such as calcifications, masses, asymmetry and carcinomas unlike existing research

work. The authors are encouraged to consider the following issues to further improve the quality of the paper.

1.The language expression needs to be more concise. There are many grammatical mistakes in the paper, so I suggest author revising revise them.

2.Data pre-processing, Results and Performance Evaluation,

The Development of the Enhanced CNN Model should not too much describe the previous work, but focus on the work of this paper.

3.The Related Work part mentioned that 30% test set and 70% training set were carried out, and after combining CBIS-DDSM Dataset, It is also divided into 60% training set and 40% test set. The Results and Performance Evaluation section also mentions having a validation set, How exactly did you divide the data set, please explain in detail.

4.The format of the table needs to be adjusted so that the words are on the same line as possible for easy reading.

5.Formula labels are duplicated and there are two different formulas (5).  Please check carefully

6.The reference is suggested to be in a unified format, and the number of reference 8 is wrong.

7.Please redesign pictures 4, 5 and 6 to ensure the sharpness of the pictures

8.Figure 5 should not take a random output as the result, but should look at the overall accuracy and the change of loss.

9.Try to keep the form to one page.

10.The paper should refer to the latest research work.

A Review of Deep-Learning-Based Medical Image Segmentation Methods, Sustainability, 2021, 13(3), 1224.

An image enhancement algorithm of video surveillance scene based on deep learning, IET Image Processing, 2021, https://doi.org/10.1049/ipr2.12286

6. PLOS authors have the option to publish the peer review history of their article (what does this mean?). If published, this will include your full peer review and any attached files.

Reviewer #1: No

Reviewer #2: **Yes: **Shuai Liu

---

## [Author Response · Author response to Decision Letter 0]

22 Jul 2021

First of all, I would like to thank you for the reviews. The comments/ suggestions from the reviewers were valuable and has allowed us to improve our paper. 

I have attached a file "Response to Reviewer- Kindly access same"

Thank you

---

## [Decision Letter · Decision Letter 1]

9 Aug 2021

Multi- class classification of breast cancer abnormalities using Deep convolutional neural network (CNN)

PONE-D-21-18653R1

Dear Dr. Heenaye- Mamode Khan,

We’re pleased to inform you that your manuscript has been judged scientifically suitable for publication and will be formally accepted for publication once it meets all outstanding technical requirements.

Kind regards,

Gulistan Raja

Academic Editor

PLOS ONE

Additional Editor Comments (optional):

Reviewers' comments:

Reviewer's Responses to Questions

**Comments to the Author**

1. If the authors have adequately addressed your comments raised in a previous round of review and you feel that this manuscript is now acceptable for publication, you may indicate that here to bypass the “Comments to the Author” section, enter your conflict of interest statement in the “Confidential to Editor” section, and submit your "Accept" recommendation.

Reviewer #1: All comments have been addressed

Reviewer #2: All comments have been addressed

2. Is the manuscript technically sound, and do the data support the conclusions?

Reviewer #1: Yes

Reviewer #2: Yes

3. Has the statistical analysis been performed appropriately and rigorously? 

Reviewer #1: Yes

Reviewer #2: Yes

4. Have the authors made all data underlying the findings in their manuscript fully available?

Reviewer #1: Yes

Reviewer #2: Yes

5. Is the manuscript presented in an intelligible fashion and written in standard English?

Reviewer #1: Yes

Reviewer #2: Yes

6. Review Comments to the Author

Reviewer #1: Authors have addressed all the comments given. The following changes are incorporated by the authors:

Research gap has been mentioned in the related work.

Authors have given clarification for the preprocessing of experimental images. The comparison of ResNet50 and CNN techniques is included in the results section.

Future work has been mentioned.

I appreciate and congratulate all the authors for their efforts in preparing this research article and wish them all the best for their future work.

The paper may be accepted in the current state.

Reviewer #2: The revision seems good enoguh. I have no other comments since all previous comments from me were addressed.

7. PLOS authors have the option to publish the peer review history of their article (what does this mean?). If published, this will include your full peer review and any attached files.

Reviewer #1: No

Reviewer #2: No

---

## [Editor Report · Acceptance letter]

16 Aug 2021

PONE-D-21-18653R1 

Multi- class classification of breast cancer abnormalities using Deep Convolutional Neural Network (CNN) 

Dear Dr. Heenaye- Mamode Khan:

I'm pleased to inform you that your manuscript has been deemed suitable for publication in PLOS ONE. Congratulations! Your manuscript is now with our production department. 

Kind regards, 

on behalf of

Dr. Gulistan Raja 

Academic Editor

PLOS ONE